# End-to-End Breast Mass Classification on Digital Breast Tomosynthesis

## Abstract

Automatic classification of the masses in digital breast tomosynthesis(DBT) is still a big challenge and play a crucial role to assist radiologists for accurate diagnosis.In this paper,we develop a End-to-End multi-scale multi-level features fusion Network (EMMFFN) model for breast mass classication using DBT. Three multifaceted representations of the breast mass (gross mass, overview, and mass background) are extracted from the ROIs and then, fed into the EMMFFN model at the same time to generated three sets of feature map.The three feature maps are finally fused at the feature level to generate the final prediction.Our result show that the EMMFFN model achieves a breast mass classification AUC of 85.09%,which was superior to the single submodel who only use one aspect of patch.

**Keywords:** Breast; Mammography; Deep Learning; Neural Network; Classification

# 1.Introduction

The 2018 Global Cancer Statistics show that Breast Cancer accounts for approximately 11.6% for 18.1 million new cancer cases and 6.6% of cancer-related deaths each year[1].Early detection with correct diagnosis is extremely important to increase the survival rate.Digital breast tomosynthesis(DBT) is a promising new breast imaging reconstructed by low-dose projections from limited scanning angles. Due to its quasi-3D tomosynthesis chatacteristics,it can reduce or eliminate the effects of tissue overlap and structural noise,which can effectively improve the accuracy of diagnosis and screening.

Computer-aided diagnosis (CAD) has been actively investigated in recent decades as an alternative and complementary approach to conventional reading by radiologists[2]. Traditional CAD methods usually require hand-crafted features extracted from mammography images for further discrimination modelling on classic classifiers for breast disease diagnosis such as breast density classification[3-6] and breast cancer classification[7-10].M.Alhelou et al.[6] propose a image preprocessing density mammogram images using a set of robust texture and edge related features combined with an SVM classifier.Narain Ponraj et al.[9]compares the approach to classify the mammogram based on the features extracted using local binary pattern and local gradient pattern with their histograms.

More recently,the development of the deep convolution neural network (DCNN) is efficiently and conveniently method,which integrates feature extraction and classification in a unified framework. Successful applications of the DCNN have been reported in breast classification. For example, Ravi K.Samala et al.[11] use a layered pathway evolution method,which through two transfer learning to compress a DCNN for classification of masses in DBT.Neeraj Dhungel et al.[12] propose a multi-view deep residual neural network,which have six input images,includeing mammogram views and binary segmentation maps of the masses and micro-calcifications,to fully automated classification of mammograms.However,since the limitations of the adequate of training medical data and lack of physical interpretation of the extracted features,using DCNN can only show suboptimal performance.

To solve the above problems,increasing researcher have tried to combine the DCNN and traditional learning methods to implemented on breast mass classification.For example, B.Mughal et al.[13] use a combination of Hat transformation with GLCM for feature extraction and use them as the input for precise classification of breast masses using back propagation neural network.S.Beura et al.[14] proposed an effective feature extraction algorithm using two dimensional discrete wavelet transform based multi-resolution analysis along with gray-level co-occurrence matrix to compute texture features and then fed them to a back propagation neural network to predict the mamogram.Further,to better avoid manual feature extraction,Xie et al.[15] proposed a multiview knowledge-based collaborative deep learning model to integrate sub-models built on nine fixed views that was decomposed into three different patches that characterize varying aspects of the target. They have claimed to obtain improved discrimination accuracies in benign/malignant lung nodule classifications.

In this paper, a End-to-End multi-scale multi-level features fusion Network (EMMFFN) model will be proposed to classify benign-malignant breast mass on DBT screening mammography.The proposed methodology construct three improved DenseNet based sub-models and fuse their feature map for further classification.

## 2.Dataset

There are totally 927 views (460 CC, 4 ML, and 463 MLO views) and 471 masses from 441 patients(age:24~90,mean 46.5) apply on current study. For each view,DBT images were acquired on a Selenia Dimensions Mammography System(Hologic Inc,Marlborough,Massachusetts,U.S) with an in-plane resolution of 0.0889 x 0.0889 mm and reconstructed at 1-mm slice spacing.

## 3.Method

We have summarized our proposed EMMFFN model in Fig 1. The model consists of three major stages: (1) extracting three types of patches, including the gross mass (GM) patches, mass background (MB) patches, and overview (OA) patches on 2D DBT mass slices; (2) construction a EMMFFM model for patch-based mass classification,and (3) classifying each mass based on the labels of its patchs.

We select a square ROI encapsulating the mass and 50 pixels above and below it to represent the OA patch. To characterize the nodule's voxel value,non-nodule voxels inside the OA patch were set to 0 to get the MB patch.To generate the GM patch,nodule voxels inside the OA patch were set to 0.Then,three types of patchs will be fed into the EMMFFN model at the same time.

The proposed EMMFFN model include three submodel,which is improved by DensNet-121,consisting of a trunk part and multiple branches,where the truck part is similar with Densnet121 and the branches are extracted from different depths of the trunk DenseNet. Multi-scale and multi-level feature maps at different resolutions can be obtained by using Max pooling layers with different sizes.We change the ratio between the kernel size and stride size of the pooling,so that the pooled feature map can contain different information. Three feautre maps are finally fused via a softmax activation layer at the end of the network.

# EMMFFN

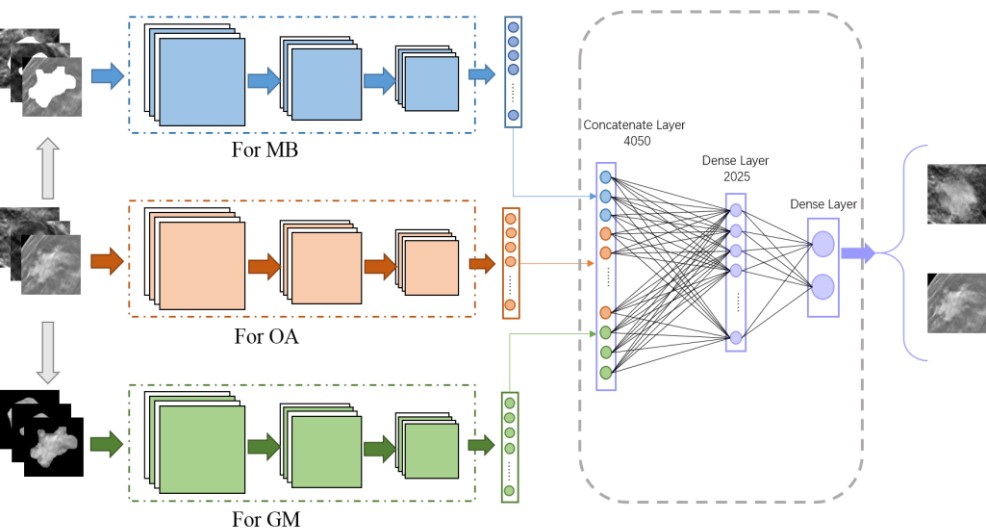

**Fig 1.** End-to-End multi-scale multi-level features fusion Network (EMMFFN)

## 4.Result

The proposed EMMFFN method was applied to our dataset 5 times independently,with 5-fold cross validation.The mean and standard deviation of obtained accuracy,sensitivity,specificity and area under the receiver operator curve (AUC),together with the performance of three single submodel,were given in Table 1. We made the following two observations: First,the GM region is apparently a more superior representation of mass characteristics in differentiating between benign and malignant breast masses, as it is foreseeable and intuitively legitimate.Second,the EMMFFN achieved higher classification performance than the single submodel in terms of AUC.

**Table 1.** Performance comparisons between EMMFFN vs. Single Submodel, with the best results marked in bold

| Models | Metrics ( %) | | | |
|---|---|---|---|---|
| | ACC | SEN | SPE | AUC |
| MB | 74.47 | **73.98** | 75.27 | 83.31 |
| OA | 74.73 | 69.11 | 77.16 | 80.39 |
| GM | **79.71** | 70.9 | **83.68** | 85.02 |
| EMMFFN | 78.26 | 72.35 | 80.86 | **85.09** |

## 5.Conclusion

In conclusion,we extensicely evaluated a novel End-to-End multi-scale multi-level features fusion Network model for breast mass classification using DBT.We used three improved DensNet121 to characterize three types patchs of breast mass,and integrate these models at the feature layer to increase the benign/malignant mass classification performance.

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
