# OpenReview forum: "End-to-End Breast Mass Classification on Digital Breast Tomosynthesis  "
_MIDL.io/2020/Conference — Submitted to MIDL 2020_

### Official Review · AnonReviewer2 · 2020-03-10
**End-to-End Breast Mass Classification on Digital Breast Tomosynthesis**

**Rating:** 2
**Confidence:** 4

**Review:**

The introduction is so long while there is less attention to their work, materials and methods. While there is a large number of patients I am wondering why the authors have not divided their dataset into training and validation sets. It is not clear for me if the images have been annotated by radiologist or by automatic algorithm. The authors have not mentioned what is the imaging technique (while I’m assuming it’s MRI). I don’t think this paper in this format is suitable for publication.

---

### Official Review · AnonReviewer4 · 2020-03-12
**End-to-End Breast Mass Classification on Digital Breast Tomosynthesis**

**Rating:** 2
**Confidence:** 5

**Review:**

The paper proposes an automatic classification of the masses in digital breast tomosynthesis (DBT) assist radiologists for accurate diagnosis. An End-to-End multi-scale multi-level features fusion Network (EMMFFN) model for breast mass classification using DBT. they extract thress multi-faceted representations of the breast mass (gross mass, overview, and mass background) from the ROIs and feed into the EMMFFN model simultaneously. The performance of the models are promising (AUC 85.09%) but lack details of model parameters, training and comparison results with existing methods.

- What best describes the contribution of this paper? Please take the paper type into consideration for the rest of your evaluation. For instance, a strong method paper should not be rejected for limited validation. Similarly, a strong validation paper should not be rejected because of lack of methodological novelty.
   O methodological development
   O validation/application paper
   O both

validation/application paper

- In 3-5 sentences, describe the key ideas, experiments, and their significance.
Multi-modality information from three types of patches is used -  gross mass (GM), mass background (MB) patches, and overview (OA) patches on 2D DBT. Performance of each individual modality is compared with a multi-modal model showcasing a superior performance of the fusion.

- What are the strengths of the paper? Clearly explain why these aspects of the paper are valuable.
The proposed EMMFFN method using three improved DensNet121 models was applied to their dataset of DBT images to characterize three types of patches of breast mass,and integrate these models at the feature layer to increase the benign/malignant mass classification performance. The model extracts three types of patches - gross mass (GM), mass background (MB) patches, and overview (OA) patches on 2D DBT mass slices for fusion into the deep learning pipeline for cancer classification.


- What are the weaknesses of the paper? Clearly explain why these aspects of the paper are weak. Please make the comments very concrete based on facts (e.g. list relevant citations if you feel the ideas are not novel) and take the paper type (method or validation paper) into account.

Multiple spelling mistakes, missing spaces after commas throughout the paper
Paper is not formatted per the author guidelines of MIDL 2020.
Fusion of features from different modalities is a quite common technique
Specific details of individual Densenet networks is missing
Authors change the ratio between the kernel size and stride size of the pooling layers so that the pooled feature map can contain different information but how it is achieved is not well understood.
Comparison with other existing approaches is missing

- What would you like the authors to address in their rebuttal? (Focus on points that might change your mind.)
Formatting of the paper per the guidelines of MIDL-2020

- List any further comments and suggestions for minor improvements or clarifications in the paper.
None

- Rating (4: Strong Accept, 3: Weak Accept, 2: Weak Reject, 1: Strong Reject)
2 Weak Reject


- Justification for rating
The proposed methods and improvements are very valuable but needs more experiments and clarifications to be validated.

Confidence
5

---

### Official Review · AnonReviewer3 · 2020-03-13
**The authors propose a model for breast mass classification using different manual patches on the given input image. Useful result. Needs work.**

**Rating:** 2
**Confidence:** 4

**Review:**

The authors suggest to use three different types of global summary patches to drive the breast mass classification, rather than using the input image directly. This is sensible. Sometimes extracting weak-label type information or force thresholding some input features (e.g., voxels are forced to 0 for some regions, like what is going on here) may be useful. But the paper needs improvement.

What is the patching doing that it may not be captured in the first layer of the network i.e., it patching is averaging and thresholding of voxel values, wont that be capturable in the first few layers of the network (and this also relates to the SPE values of proposal and individual models, see below)?
The presentation can be improved. What is the intuition for specifically using these three patch types? Is the output space voxel-wise? It is not clear what we are looking at here.
Do we expect the three modalities to behave differently in terms of lower layers i.e., do we need different architectures for each independent modalities (the left blocks in Figure).
Are the results reported in Table significantly different? Why is the specificity decreasing but accuracy increasing (anything w.r.t data set or imbalanced classes).

---

### Official Review · AnonReviewer1 · 2020-03-14
**Classification network needs more details**

**Rating:** 1
**Confidence:** 5

**Review:**

The manuscript presents a network in the form of an ensemble of 3 parallel DenseNet arms focusing on the gross mass (GM) patches, mass background (MB) patches, and overview (OA) patches individually for the classification into benign or malignant mass.
The manuscript lacks necessary details in implementation. The design of the network lacks necessary justification. Results shows marginal improvements compared with each individual arm.
One significant problem is the ambiguity in the determination of GM, MB, and OA which is a segmentation problem.
Overall the contribution of the manuscript is limited and the quality needs significant improvement.

---

### Meta-Review · Area_Chair1 · 2020-03-28
**MetaReview of Paper22 by AreaChair1**

**Rating:** 2

**Metareview:**

All reviewers suggest that the proposed method is a quite standard multi-modal approach, but interesting in the sense that it uses different types of input from DBT.  Nevertheless, the paper has important issues that need to be addressed.  For instance, there are implementation details missing, where some design decisions are not well justified.  In particular,  why does the paper use these three patch types?  Also, the multi-modal approach does not seem to perform significantly better than the mono-modal ones.  Another issue was the specificity result that decreases with an increasing accuracy, which is a bit odd.  To summarise, the paper shows a standard multi-modal classification that relies on new input types from DBT, but it lacks implementation details and shows results that do not seem to be relevant.  Therefore, I agree with the reject rating by the reviewers.

**Paper Type:**

validation/application paper

---

### Decision · Program_Chairs · 2020-04-11

Reject